# Effects of Dietary Supplementation of Stimbiotics to Sows on Lactation Performance, Immune Function, and Anti-Inflammatory and Antioxidant Capacities during Late Gestation and Lactation

**DOI:** 10.3390/vetsci11020053

**Published:** 2024-01-26

**Authors:** Jing Li, Wen-Ning Chen, Wen-Juan Sun, Gustavo Cordero, Shah Hasan, Valentino Bontempo, Jun-Feng Xiao, Yan-Pin Li, Yu Pi, Xi-Long Li, Xian-Ren Jiang

**Affiliations:** 1Key Laboratory of Feed Biotechnology of the Ministry of Agriculture and Rural Affairs, Institute of Feed Research, Chinese Academy of Agricultural Sciences, Beijing 100081, China; lijingbj_87@126.com (J.L.); 82101215363@caas.cn (W.-N.C.); sunwenjuan@caas.cn (W.-J.S.); liyanpin@caas.cn (Y.-P.L.); piyu@caas.cn (Y.P.); lixilong@caas.cn (X.-L.L.); 2AB Vista, Marlborough SN8 4AN, UK; gustavo.cordero@abvista.com (G.C.); shah.hasan@abvista.com (S.H.); 3Department of Veterinary Medicine and Animal Science (DIVAS), University of Milan, 26900 Lodi, Italy; valentino.bontempo@unimi.it; 4Key Laboratory of Swine Nutrition and Feed Science of Fujian Province, Aonong Group, Zhangzhou 363000, China; junfeng.xiao@aonong.com.cn

**Keywords:** stimbiotic, sow, colostrum, milk, immunoglobulin, plasma, feed additive

## Abstract

**Simple Summary:**

In recent years, stimbiotic supplementation was found to be an innovative nutritional strategy to enhance animal performance, gut health, and immune function. Stimbiotics work by extracting nutrients from dietary fiber, increasing fiber utilization in the distal intestine, and promoting the production of short-chain fatty acids (SCFAs). This study aimed to evaluate the effect of a stimbiotic on sow performance, immune function, and anti-inflammatory and antioxidant capacities during late gestation and lactation. In general, stimbiotic supplementation increased the body weight gain of the weaning piglet during the lactation period. Moreover, stimbiotic supplementation may improve the immune function of lactating sows and their piglets by increasing the plasma concentration of IgM at farrowing or the plasma concentrations of IgG and IgA at weaning. Furthermore, stimbiotic supplementation increased the concentration of IgM in the colostrum and of IgM and IgA in the milk at d 14 of lactation. The supplementation of the stimbiotic tended to reduce plasma concentrations of malondialdehyde (MDA) in sows during lactation. These findings indicated that stimbiotic supplementation could be a potential feed additive for improving the reproductive performance and immunity of sows during lactation.

**Abstract:**

Stimbiotic supplementation may provide an innovative feed additive solution to accelerate the proliferation of beneficial fiber-degrading bacteria in the distal intestine and the utilization of dietary fiber. Optimal utilization of dietary fiber has multiple benefits for gut health and nutrient utilization. This study was conducted to evaluate the late gestation and lactation performance, the plasma, colostrum, and milk immunoglobulin (IgA, IgG, and IgM) concentrations, and the anti-inflammatory and antioxidant biomarkers in plasma of sows fed with or without a stimbiotic during the late gestation and lactation phase. A total of 40 sows were allocated to two treatment groups: control (CT) with no supplementation or 100 mg/kg stimbiotic (VP), with 20 sows per treatment. Sows were fed the treatment diets from d 85 of gestation to d 28 of lactation. In the results, the average daily weight gain of piglets during lactation was greater from sows fed in the VP group compared to that in the CT group (*p* < 0.05). The plasma concentrations of IgM at farrowing and IgG at weaning of the sows fed the diet with the stimbiotic supplementation were much higher than those in the CT sows (*p* < 0.05), respectively. In addition, the dietary stimbiotic increased the concentrations of IgM in the colostrum and of IgA and IgM in the milk at d 14 of lactation (*p* < 0.05). Plasma concentrations of malondialdehyde (MDA) on d 0 and d 28 of lactation tended to be lower in sows fed the VP diets compared with those of the sows fed the CT diets. Thus, our study indicated that stimbiotic supplementation could improve the daily weight gain of piglets and the immune function of sows in lactation.

## 1. Introduction

In recent years, stimbiotics, a combination of xylanase and xylo-oligosaccharides, have been developed as innovative feed additive solutions to improve gut health in farm animals such as pigs and broilers. Stimbiotics are expected to improve fiber fermentation in the lower gut without acting as a direct substrate for such fermentation [1]. Recent research has reported that stimbiotic supplementation enhanced animals’ growth rates and gut health by extracting extra energy from dietary fiber, increasing the utilization of the undigested dietary fiber by promoting fiber-degrading bacterium, breaking down the complex fiber structure into a lower degree of polymerization (DP) fiber, and eventually promoting the production of short-chain fatty acids (SCFAs) in the animal’s distal intestine [2,3]. It is well-known that SCFAs, as a main energy source of intestinal epithelial cells (IECs), play an important role in the improvement of the gut’s integrity and immune function. Short-chain fatty acids may also inhibit pathogenic bacteria and reduce the inflammatory response and intestinal diseases, which provide multiple beneficial effects on the host’s health and performance [4,5]. Furthermore, previous authors have reported that stimbiotic supplementation was able to reduce the growth of pathogenic bacteria and the inflammatory response, as well as increase the gut integrity, immunoglobulin levels, and insulin-like growth factor 1 (IGF-1) levels when the animals were facing several challenges in their feeding environment [6,7,8,9].

In swine production, it is well-known that the reproduction performance of sows, including litter size, litter weight, number of piglets born alive, and milk quality, are closely associated with the subsequent weaned piglets’ growth performance and the economic efficiency of the pig farm [10,11]. However, in most instances, sows are susceptible to the challenge of oxidative stress, as well as stresses from farrowing and milk production during late gestation and lactation, all of which are closely associated with the sow’s reproductive and lactation performance. It is widely known that oxidative stress could lead to a decline in immune function. Excessive reactive oxygen species (ROS) production results in damage to the proteins, lipids, and DNA, and eventually may induce cell metabolism disorders and inflammation, all of which can negatively impact animal performance [12,13]. In recent years, numerous studies have proven that the oxidative stress status of sows could be alleviated through a variety of nutrient interventions, such as antioxidants, including vitamins and minerals, plant extracts, prebiotics, and probiotics [14,15,16,17].

In a previous study, it was observed that stimbiotic supplementation significantly increased the body weight gain and immune function of weaned piglets by accelerating the growth of beneficial fiber-degrading bacteria and fiber fermentation in the hind gut [9]. Moreover, recent research suggested that stimbiotic supplementation could improve sow productivity by increasing the fermentative capacity of dietary fiber in gestating sows [18]. However, the effect of stimbiotics on reproductive performance, especially on oxidative stress and the immune response of sows, has not been well-studied. Therefore, the objective of the present study was to further evaluate the effect of stimbiotic supplementation on the lactation performance, anti-inflammatory capacity, and antioxidative capacity of sows during the late gestation and lactation phases.

## 2. Materials and Methods

### 2.1. Ethic Statement

The animal experimental procedures in the study were conducted according to the Institute Animal Care and Use Committee of the Institute of Feed Research of the Chinese Academy of Agricultural Sciences (IFR-CAAS20220320).

### 2.2. Animals and Experimental Design

A total of 40 sows (Yorkshire × Landrace) from 3rd to 5th parturition were selected at 85 days of gestation and randomly divided into two groups (n = 20) based on their health and body weight (251.7 ± 5.6 kg). The control group (CT) was fed the basal diet and the VistaPros (VP) group was fed the basal diet supplemented with 100 mg/kg VistaPros (a stimbiotic product that was supported by AB Vista Asia Pte., Ltd., Singapore). The basal diets for sows in gestation and lactation were formulated by following the National Research Council’s (NRC, 2012) nutritional requirements (Table 1).

From d 85 to 107 of gestation, sows were housed in the same room with a separate gestation cage (2.2 m × 0.70 m) for each sow and fed 3 kg experimental diet per day in divided portions (07:00 a.m. and 3:00 p.m.). On day 108, the sows were transferred to separate farrowing crates (2.2 m × 1.8 m). On the day of farrowing, sows were not fed, and in the first 5 days after farrowing, sows were fed progressively 4 times per day (07:00 a.m., 11:00 a.m., 3:00 p.m., and 6:00 p.m.), starting from 1.0 kg/d and increasing by 1.0 kg per d; afterwards, all sows were fed ad libitum until weaning. The feed intake of the sows throughout the lactation period was recorded and used to calculate the average daily feed intake during lactation. Cross-fostering was performed within the group during the first 24 h after farrowing, with 10–12 piglets per sow. All of sows had access to water freely. The average temperature of the gestation room was 22 °C and 25 °C in the farrowing room, and heat lamps (250 W) were used to provide extra heat for the piglets.

### 2.3. Measurements and Samples Collection

Sow body weight and backfat thickness was measured at d 85 and d 107 of gestation and again at weaning (d 28 of lactation). Backfat thickness was measured using ultrasound equipment (Renco Lean-Meatier; Renco Corporation, Manchester, MA, USA) at 65 mm right of the dorsal midline of the last rib. At farrowing, the total number of piglets born, the number of piglets born alive, and the number of piglets born dead were recorded for each sow. The piglets were weighed within 24 h after farrowing and again at weaning (d 28 of lactation).

A 10 mL sample of the first colostrum of each sow was gathered from all the active mammary glands on the right side of the sow within one hour during farrowing. On d 14 of lactation, 20 mL samples of milk from each sow were collected approximately 30 s after sows were injected with 2 mL (equal to 10 units) of oxytocin intramuscularly in the hip (Ningbo Sansheng Biotech, Ningbo, China). The milk samples were collected into sterile tubes and immediately stored at −20 °C until further analysis. On d 0 and 28 of lactation, 10 mL of blood was collected from the ear vein of each sow, placed in heparin tubes, and centrifuged at 3000 rpm for 20 min by high-speed centrifuge (Thermo Fisher, Karlsruhe, Germany). The plasma was obtained, and samples were stored at −20 °C for later analysis.

### 2.4. Immunological and Inflammatory Indicators

Plasma, milk, and colostrum samples were further analyzed for immunological indicators, containing immunoglobulin A (IgA), immunoglobulin M (IgM), and immunoglobulin G (IgG). Plasma was also analyzed for inflammatory indicators, including interleukins-6 (IL-6), interleukins-10 (IL-10), interleukins-1β (IL-1β), and tumor necrosis factor-α (TNF-α). Immune- and inflammatory-specific ELISA kits (mlbio, Shanghai, China) for pigs were used to determine IgA, IgG, IgM, interleukin, and TNF-α levels according to the manufacturer’s instructions. Plasma was extracted using a volume of 10 μL with 40 μL of diluent mixed in and added to a 96-well plate followed by 100 μL of horseradish peroxidase and incubated at 37 °C for 1 h. After five washes, chromogen solution was added to the 96-well plate and the reaction was carried out at 37 °C for 15 min protected from light. The reaction was finally terminated by adding 50 μL of the termination solution. The absorbance was measured at 450 nm, the standard curve was generated from the concentration and optical density (OD) of serially diluted standards, and the sample concentration was calculated from the standard curve.

### 2.5. Antioxidant Capacities

The antioxidant indices from each plasma sample were measured. These included the levels of glutathione peroxidase (GSH-Px), total antioxidant capacity (T-AOC), superoxide dismutase (SOD), and malondialdehyde (MDA). Plasma GSH-Px activity was determined by the pig ELISA kit (mlbio, Shanghai, China) by following the kit instructions. Briefly, the ELISA kit measures the level of GSH-Px in each sample via a double-antibody sandwich method and detects the absorbance value at 450 nm for calculating the concentration from the standard curve. Determination of SOD, T-AOC, and MDA levels in plasma were based on the protocols of the individual ELISA kits for pigs (Liancheng Bioengineering Institute, Nanjing, China). Plasma SOD activities were determined by the water-soluble thiazole salt (WST)-1 method, and the OD value was measured at 450 nm. The concentration of MDA was determined by the 2-thiobarbituric acid (TBA) method, and the change in absorbance was recorded at 532 nm. The level of plasma T-AOC was measured using the FRAP method, and the OD value was read at 405 nm. All the OD values, at different nanometers (nm), were conducted on a microplate spectrophotometer in this experiment (BioTek, Winooski, VT, USA).

### 2.6. Statistical Analysis

Data were analyzed as a randomized complete block design using the generalized linear model (GLM) procedure of SAS 9.2. In this model, sample size was calculated by means using the standard error of the means (SEM). For the analysis of reproductive performance, each sow and the piglets per litter and pen in the CT or VP group served as an experimental replicate. For the analysis of immunological and inflammatory factors and antioxidant parameters, each sow was used as the experimental unit. All the data in this experiment were considered as significant when *p* < 0.05 and propensity was indicated when 0.05 < *p* < 0.10.

## 3. Results

### 3.1. Lactation Performance

As shown in Table 2, no differences in sows’ body weight and average backfat thickness at d 85, d 107, or at weaning were observed between the CT and VP groups (*p* > 0.05). Furthermore, there was no difference in average daily feed intake of the sows fed the CT and VP diets during the lactation period (*p* > 0.05).

There were no differences in the litter weight or body weight of piglets at farrowing or weaning observed in the two groups (*p* > 0.05) (Table 3). However, the average daily gain of piglets in the VP group was significantly higher than that in the CT group during the lactation period (*p* = 0.021) (Table 3).

### 3.2. Immunoglobulin Concentration in Plasma and Milk

The plasma concentrations of IgA, IgG, and IgM at d 0 and d 28 of lactation in the CT and VP sows are presented in Table 4. The plasma concentrations of all immunoglobulins in the VP group were numerically higher than those in the CT group. Meanwhile, the plasma concentrations of IgM at d 0 and IgG at d 28 of lactation were significantly increased in the VP group (*p* = 0.021, *p* = 0.022), and the IgA levels at d 28 of lactation tended to be greater in the VP group (*p* = 0.087) compared with sows fed in the CT group.

In addition, the concentrations of IgM in colostrum (*p* = 0.032) and in d 14 milk (*p* < 0.01) and of IgA in d 14 milk (*p* < 0.01) were remarkably increased in the VP sows compared to those in the CT sows (Table 5). The concentrations of IgG in the d 14 milk of VP sows tended to be increased *(p* = 0.094) compared with those of sows fed in the CT group. These results indicate that higher concentrations of immunoglobulins were found in VP sows’ colostrum and subsequent milk.

### 3.3. Plasma Inflammatory Cytokines

The plasma concentrations of IL-10, IL-1β, IL-6, and TNF-α at d 0 and d 28 of lactation in the CT and VP groups are presented in Table 6. The results demonstrate that there were no differences in the inflammatory indicators between the two groups (*p* > 0.05). In addition, the plasma concentrations of IL-10 and IL-1β were lower and IL-6 and TNF-α were higher at d 28 of lactation when compared to those at d 0 of lactation numerically, which indicated higher inflammatory responses at d 28 of lactation.

### 3.4. Plasma Antioxidant Capacities

The plasma concentrations of T-AOC, SOD, GSH-PX, and MDA at farrowing and weaning (d 28 of lactation) in the CT and VP groups are presented in Table 7. These results demonstrate that no differences in the oxidative stress indicators between the CT and VP groups were observed at farrowing or weaning (*p* > 0.05). However, the plasma concentrations of MDA at farrowing and weaning in the VP group tended to be lower than those in the CT group (*p* = 0.080; *p* = 0.080).

## 4. Discussion

In the present study, no difference was found in the body weight, backfat thickness, or feed intake of sows during late gestation and lactation when fed diets with or without stimbiotics. There was also no effect of stimbiotic supplementation on the litter size or litter birth weight of piglets at farrowing. These findings were not unexpected and may be the result of the short-term experimental period of stimbiotic supplementation and the management of the nutrient intake of sows during the whole gestation, which is consistent with the previous studies for other feed additives during late gestation and lactation in sows [19]. Furthermore, many researchers have reported that the litter size and litter birth weight of piglets are closely associated with early embryonic development, especially embryonic survival before d 30 of gestation, and nutrient supplementation from the placenta of sows during gestation [20,21,22,23,24]. However, an increased body weight gain of piglets from sows fed VP during lactation was observed and was significantly higher than that of piglets from the CT sows. These results may demonstrate that stimbiotics provide more energy to the piglet from the colostrum and milk of sows fed with stimbiotics. In fact, some recent research has reported stimbiotics enhanced the fermentation and utilization of fiber, producing SCFAs in the distal intestine of pigs and poultry. Of the SCFAs, butyric acid is considered as an important energy source for the intestinal epithelium. In this regard, the extra energy from the sow to the piglet for weight gain was likely from improved dietary fiber utilization with the supplementation of the stimbiotic [7,25,26,27,28].

To our knowledge, SCFAs, including acetate, propionate, and butyrate, produced by hindgut bacterium from fermented indigestible carbohydrates (mainly fiber), promote intestinal barrier functionality. Short-chain fatty acids are also shown to regulate gut-derived hormone secretion (e.g., GLP-1, GLP-2, and PYY) and immunoglobulin levels by the GPR (G-protein-coupled receptor) 41, GPR43, and GPR109A signaling pathways. Finally, SCFAs are reported to reduce the inflammatory response through inducing Treg cell differentiation or inhibiting the gene expression of pro-inflammatory cytokines, such as IL-1β, IL-6, and TNF-α, via the TLR/NF-κB signaling pathway [29,30,31,32,33].

In the current study, stimbiotic supplementation significantly increased IgA, IgG, and IgM antibodies in the plasma, colostrum, and d 14 milk of sows. These are the main immunoglobulins of lactating sows. At birth, piglet immunity is only derived from colostrum, and it is ideal for the colostrum and milk of sows to have a higher immunoglobulin concentration to strengthen maternal immune protection and infantile anti-infection capacity [34,35,36]. In particular, IgA is the immunoglobulin of the largest proportion in milk. In the current study, on d 14, the concentration of IgA in the milk of VP sows was significantly higher than that in CT sows, which indicates that stimbiotics could promote the secretion of IgA in milk from maternal plasma [37]. A previous report further demonstrated that the microbiota metabolite acetate could modulate intestinal IgA production by activating G-protein-coupled signaling pathways [38], which probably explains the higher concentration of IgA from stimbiotic supplementation in the current study.

It is well-known that the late gestation and lactation periods are critical periods in the reproductive cycle of sows. The physiological processes of sows during this time are undergoing dynamic changes with a high metabolic requirement to provide nutrients through the placenta to facilitate the fast growth rate of the fetus and prepare for farrowing and the production of milk, all of which can be accompanied by inflammation and the generation of a series of oxidative stressors [12,39]. Thus, the current study further evaluated the anti-inflammatory and antioxidant abilities of sows fed with the supplement of stimbiotic. In this case, there were no differences observed for the plasma IL-1β, IL-6, and TNF-α levels between the VP and CT groups. However, previous authors have demonstrated that stimbiotic supplementation could significantly down-regulate plasma TNF-α in piglets at d 35 post-weaning when housed in poor sanitary conditions by increasing the production of SCFAs from fiber fermentation [6]. In addition, in the current study, the concentrations of MDA in the plasma of VP-fed sows tended to decrease at farrowing and at weaning (d 28 of lactation). Plasma MDA is a known key marker of oxidative stress and is reported to be decreased by adding high concentrations of soluble fiber into the diets of sows during gestation and lactation [40], which may be correlated with gut microbiota, fermentation, and its metabolite SCFAs [41]. In addition, some studies have suggested that SCFAs produced by gut bacteria fermentation, especially butyrate and propionate, could elevate cellular redox homeostasis and inhibit oxidative stress via the Nrf2 signaling pathway and lipid metabolism [42,43,44]. Taken together, the previous studies and current results indicate that stimbiotic supplementation may reduce the inflammatory response and oxidative stress of sows during gestation and lactation by promoting the beneficial fiber-degrading bacteria to utilize dietary fiber and produce SCFAs. This results in a measured increase in plasma, colostrum, and milk immunoglobulins for use by the piglets and a decrease in MDA content in sows fed the stimbiotic.

## 5. Conclusions

In summary, the supplementation of a stimbiotic into sow diets during the late gestation and lactation periods could increase the weight gain of piglets during lactation and enhance the levels of immunoglobulins in the plasma, colostrum, and milk of lactating sows. In addition, the stimbiotic also tended to reduce the oxidative stress of lactating sows, which indicates that stimbiotics could be used as a feed additive in sow diets to improve sows’ resilience to stressors and piglets’ performance during the lactation period.

## Figures and Tables

**Table 1 vetsci-11-00053-t001:** Ingredients and calculated nutrient composition of the basal diet (as fed basis, %).

Items	Gestation	Lactation
Ingredients		
Corn	64.65	62.94
Soybean meal, 43%	16.00	18.00
Fish meal	-	3.00
Extruded soybean	-	8.00
Rice bran	8.00	-
Bran	7.00	4.00
Soybean oil	1.40	1.00
Dicalcium phosphate	0.20	0.90
Limestone (CaCO_3_)	1.60	1.05
L-lysine, 70%	0.15	0.11
Vitamin and mineral premixes ^1^	1.00	1.00
Total	100.00	100.00
Analyzed nutrient content		
Protein, %	14.36	18.10
Calcium, %	0.78	1.15
Phosphorus, %	0.74	0.71
Fat, %	5.57	5.99
Calculated nutrient content		
NE, kcal/kg	2470	2508
Protein, %	14.50	18.50
Calcium, %	0.70	0.76
Phosphorus, %	0.55	0.65
NDF, %	11.57	9.30
ADF, %	4.43	3.70
Lysine, %	0.65	0.90
Methionine, %	0.20	0.24
Threonine, %	0.43	0.56
Tryptophan, %	0.14	0.18

Note: ^1^ The premix contained the vitamins and trace minerals, the contents per kilogram of diet as follows: Cu, 10 mg; Fe, 80 mg; Zn, 100 mg; I, 0.3 mg; Mn, 25 mg; Se, 0.3 mg; 25,000 IU vitamin A; 50 IU vitamin E; 5000 IU vitamin D3; 2.5 mg vitamin K; 0.2 mg biotin; 1 mg vitamin B1; 8 mg vitamin B2; 3 mg vitamin B6; 0.020 mg vitamin B12; 15 mg niacin; 12.5 mg pantothenic acid; 1.50 mg folacin.

**Table 2 vetsci-11-00053-t002:** Effect of dietary VP supplementation on body weight, backfat thickness, and feed intake of sows during late gestation and lactation period.

Items	CT	VP	SEM	*p*-Value
Number of sows	20	20		
Body weight of sows, kg				
D 85 of gestation	251.3	252.0	5.6	0.932
D 107 of gestation	268.4	270.5	5.5	0.811
At weaning	235.5	232.1	6.1	0.709
Average backfat thickness, mm				
D 85 of gestation	15.91	16.62	0.32	0.165
D 110 of gestation	17.18	17.11	0.44	0.919
At weaning	15.34	14.52	0.48	0.221
Average daily feed intake/sows, kg				
During lactation	5.26	5.01	0.23	0.289

CT = control group with the basal diet; VP = VistaPros group with the supplementation of 100 mg/kg VistaPros in the basal diet.

**Table 3 vetsci-11-00053-t003:** Effect of dietary VP supplementation on the reproduction performance of sows during the late gestation and lactation period.

Items	CT	VP	SEM	*p*-Value
Number of sows	20	20		
At farrowing				
Total born	14.85	15.25	1.00	0.739
Born alive	12.60	12.10	0.80	0.655
Average litter weight, kg	18.17	16.06	0.86	0.117
Average body weight, kg	1.48	1.42	0.07	0.523
During lactation				
Average lactation days	27.95	27.85	0.53	0.911
ADG of piglets, g	227	249	9	0.021
At weaning				
Average litter size	10.60	9.80	0.42	0.159
Average litter weight, kg	83.42	80.39	4.04	0.608
Average body weight, kg	7.86	8.33	0.29	0.215

ADG = average daily gain; CT = control group with the basal diet; VP = VistaPros group with the supplementation of 100 mg/kg VistaPros in the basal diet.

**Table 4 vetsci-11-00053-t004:** Effect of dietary VP supplementation on plasma immunoglobulins of sows during the lactation period.

Items	CT	VP	SEM	*p*-Value
IgA, μg/mL				
At farrowing	1054	1058	25	0.915
At weaning	863	946	30	0.087
IgG, mg/mL				
At farrowing	29.02	29.92	0.61	0.321
At weaning	30.87	35.27	1.16	0.022
IgM, mg/mL				
At farrowing	4.88	5.16	0.08	0.021
At weaning	4.32	4.37	0.12	0.765

CT = control group with the basal diet; VP = VistaPros group with the supplementation of 100 mg/kg VistaPros in the basal diet.

**Table 5 vetsci-11-00053-t005:** Effect of dietary VP supplementation on immunological performance of sow’s milk quality during the lactation period.

Items	CT	VP	SEM	*p*-Value
IgA, μg/mL				
Colostrum	1969	1910	36	0.264
D14 after farrowing	1044	1161	19	<0.01
IgG, mg/mL				
Colostrum	45.63	48.71	1.11	0.094
D14 after farrowing	32.40	32.56	0.49	0.825
IgM, mg/mL				
Colostrum	8.12	9.03	0.25	0.032
D14 after farrowing	5.19	5.65	0.10	<0.01

CT = control group with the basal diet; VP = VistaPros group with the supplementation of 100 mg/kg VistaPros in the basal diet.

**Table 6 vetsci-11-00053-t006:** Effect of dietary VP supplementation on inflammatory response of sows during the lactation period.

Items	CT	VP	SEM	*p*-Value
IL-10, pg/mL				
At farrowing	153	155	4	0.677
At weaning	114	122	4	0.189
IL-1β, pg/mL				
At farrowing	548	571	11	0.178
At weaning	475	444	17	0.229
IL-6, pg/mL				
At farrowing	618	619	12	0.977
At weaning	727	719	14	0.697
TNF-α, pg/mL				
At farrowing	122	124	2	0.602
At weaning	152	144	8	0.476

CT = control group with the basal diet; VP = VistaPros group with the supplementation of 100 mg/kg VistaPros in the basal diet.

**Table 7 vetsci-11-00053-t007:** Effect of dietary VP supplementation on anti-oxidation capacity of sows during the lactation period.

Items	CT	VP	SEM	*p*-Value
T-AOC, mmol/mg				
At farrowing	0.050	0.061	0.004	0.112
At weaning	0.055	0.057	0.004	0.751
SOD, U/mL				
At farrowing	16.27	16.52	0.92	0.853
At weaning	12.17	12.67	0.33	0.295
GSH-Px, U/mL				
At farrowing	135	135	6	0.946
At weaning	126	148	12	0.233
MDA, nmol/mL				
At farrowing	4.00	3.56	0.16	0.080
At weaning	4.40	3.86	0.21	0.080

CT = control group with the basal diet; VP = VistaPros group with the supplementation of 100 mg/kg VistaPros in the basal diet.

## Data Availability

Datasets generated during and/or analyzed during the current study are available from the corresponding author on reasonable request.

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
