# Peer review of "Effects of Dietary Supplementation of Stimbiotics to Sows on Lactation Performance, Immune Function, and Anti-Inflammatory and Antioxidant Capacities during Late Gestation and Lactation"

_vetsci, 2024, doi:10.3390/vetsci11020053_

Round 1

Reviewer 1 Report

Comments and Suggestions for Authors   The manuscript entitled "Effects of dietary supplementation of stimbiotic to sows on lactation performance, immune function, anti-inflammatory and antioxidant capacities during late gestation and lactation" describe the effect of stimiotic on the sow in the late gestation and during the peripartal period. This topic is of major interested for the swine industry. However, before publication, the manuscript need major revisions.

Line 27: please add feed additive

Line 28-29: Please be more specific: eg. immunoglobulin content of colostrum/milk,

Line 32: Please specify this sentence “All the sows were fed from d 85 gestation to d 28 lactation”. It is necessary to feed all sows. Please add the amount of the feed additive.

Line 33-39: Please add the values of the examined parameter, so that the reader of the abstract see the total numbers and not only if it was significant.

Line 38: Please add weight: Daily weight gain

Introduction

Line 49: Two references are not a number of researches.

Line 56- 61: Please rewrite the sentence, because it is too long and also the statement is not clear.

 Material and Methods

Line 106: Can you please verify at which time points the back fat measurement was conducted

Line 111:  Please describe in detail how you received 20-mL sample of the milk of each sow on d 14? Only manual palpation is not possible, since the milk can just be received in only some seconds.

Line 114-115: Please be more specific. What is the neck vein? It is not possible to receive 10ml out of the ear vein.

Line 118: Which tissue were your analyzing for the immunoglobulin content? Please describe that it is from the blood.

Please give information on the parity of the sows?

Do you have data regarding the farrowing process such as birth duration, dystocia,…?

Furthermore, do you have data on the puerperium, such as feed intake, body temperature, vaginal discharge? This could also prove the effect on the immunsystem.

Please add the sample size calculation to this section.

Results

Line 157 shown instead of showed

Discussion

Please also discuss the limitations of the study. Sample size, observer bias   Comments on the Quality of English Language

Please check the manuscript with an native speaker.

Author Response

 Thank you very much for the great comments that aim to improve the quality of our manuscript. We have modified the manuscript according to your comments and the update details were as follows:

Comments 1: Line 27: please add feed additive

Response 1: We have added “feed additive” in Line 27

Comments 2: Line 28-29: Please be more specific: eg. immunoglobulin content of colostrum/milk,

Response 2: We have added immunoglobulin level of plasma and colostrum/milk, anti-inflammatory and anti-oxidant indicators in plasma of sows in Line 28-29.

Comments 3: Line 32: Please specify this sentence “All the sows were fed from d 85 gestation to d 28 lactation”. It is necessary to feed all sows. Please add the amount of the feed additive.

Response 3: We have added the amount of feed additive in the content. “All the sows were fed the diet with or without 100g/t stimbiotic from d 85 gestation to d 28 lactation.”

Comments 4: Line 33-39: Please add the values of the examined parameter, so that the reader of the abstract see the total numbers and not only if it was significant.

Response 4: We totally agreed with you and want to add the values of examined parameters to make it clearer, but unfortunately, it’s rather difficult to include the values as the limitation number of words in abstract as the requirement of the journal.

Comments 5: Line 38: Please add weight: Daily weight gain

 Response 5: We have already changed in Line 38.

Introduction

Comments 6: Line 49: Two references are not a number of researches.

 Response 6: We have changed “a number of researches” to “some researches” in Line 38.

Comments 7: Line 56- 61: Please rewrite the sentence, because it is too long and also the statement is not clear.

Response 7: We have re-described the sentence to make it more clearly in Line 56-61.

 Material and Methods

Comments 8: Line 106: Can you please verify at which time points the back fat measurement was conducted

Response 8: We have added “and at the same time points” for back fact measurement was conducted in Line 106.

Comments 9: Line 111:  Please describe in detail how you received 20-mL sample of the milk of each sow on d 14? Only manual palpation is not possible, since the milk can just be received in only some seconds.

Response 9: Thank you for the great comments. It’s impossible to receive 20 mL milk if only following manual palpation. Actually we injected 2 mL of oxytocin for the quick collection milk. And we have added the more information in Line 111 ( On d 14 of lactation, a 20-mL sample of the milk from each sow was collected approximately 30s after sows were injected with 2 mL of oxytocin by intramuscularly in the hip (Ningbo Sansheng Biotech, Zhejiang, China).

Comments 10: Line 114-115: Please be more specific. What is the neck vein? It is not possible to receive 10ml out of the ear vein.

Response 10: We have corrected the description in Line 114-115. We only received 10 mL blood samples from ear vein. And as we collected lots of times from sows at different phases, it’s easy to receive 10 mL blood by ear vein. 

Comments 11: Line 118: Which tissue were your analyzing for the immunoglobulin content? Please describe that it is from the blood.

Response 11: We have added more information in the 2.4 part. It was revised to the sentences as follows: “We collected 10 mL blood samples from each sow at farrowing and at weaning to analysis immunological indicators, including immunoglobulin A (IgA), immuno-globulin G (IgG), and immunoglobulin M (IgM) and also measured inflammatory in-dicators of plasma, including interleukins -6 (IL-6), interleukins -10 (IL-10), interleu-kins -1β (IL-1β) and tumor necrosis factor -α (TNF-α). For the colostrum and milk samples, we only detected immunological indicators.”

Comments 12: Please give information on the parity of the sows?

Response 12: We have added the parity of the sows in the 2.2 section animals and experimental design. “A total of 40 healthy sows (Yorkshire × Landrace) from 3rd to 5th parturition were selected at 85 days of gestation and randomly divided into two groups.”

Comments 13: Do you have data regarding the farrowing process such as birth duration, dystocia,?

Response 13: Indeed we didn’t record these exact information, but for the farrowing process, all the sows were in the regular status, there’s no dystocia.

Comments 14: Furthermore, do you have data on the puerperium, such as feed intake, body temperature, vaginal discharge? This could also prove the effect on the immunsystem.

Response 14: Thank you for the great comments. All the sows during the experimental period, there were no abortion or abnormal occurrence, so we didn’t collect the data of body temperature or vaginal discharge. In addition, for the feed intake we have added more information in Line 95-97.

Comments 15: Please add the sample size calculation to this section.

Response 15: We have added the samples sized information in the 2.6 section statistical analysis. “ In this model, sample size was calculated by means using the standard error of the means (SEM). ”

Results

Comments 16: Line 157 shown instead of showed

 Response 16: It was modified to shown in Line 157

Discussion

Comments 17: Please also discuss the limitations of the study. Sample size, observer

 Response 17: We have added some information in discussion part to make it more sense.

Comments 18: Comments on the Quality of English Language

Please check the manuscript with an native speaker.

 Response 18: Thank you for the great comments. We have asked a scientist in native English language to review this manuscript.

Reviewer 2 Report

Comments and Suggestions for Authors

In general, the article is clearly written, and the objective is defined. Material and methods are correctly detailed. Results, discussion, and conclusions are appropriate.

Table 1: note 1 should be in the table as a superscript.

Page 3, lines 104-107: It is not clear whether backfat thickness is measured at the same days as body weight. Please, express it more explicitly.

Table 3: Please, indicate the meaning of the abbreviation ADG in the table footer.

Page 8, line 246: Please, correct "previous”.

Author Response

Thank you very much for your kind comments. We have modified the manuscript according to your comments.  

The modified details were as follows:

 Comments 1: Table 1: note 1 should be in the table as a superscript.

Response 1: We have changed “note” to “Note” in Table 1.

Comments 2: Page 3, lines 104-107: It is not clear whether backfat thickness is measured at the same days as body weight. Please, express it more explicitly.

Response 2: We have added “and at the same time points” in Line 104-107 to describe more explicitly.

Comments 3: Table 3: Please, indicate the meaning of the abbreviation ADG in the table footer.

Response 3: Thank you for the correction. We have added the ADG meaning (average daily gain) in the table footer of Table 3.

Comments 4: Page 8, line 246: Please, correct "previous”.

Response 4: We have corrected the type in Line 246.

Reviewer 3 Report

Comments and Suggestions for Authors

Dear Authors,

Your manuscript titled "Effects of dietary supplementation of stimbiotic to sows on lactation performance, immune function, anti-inflammatory and antioxidant capacities during late gestation and lactation" is suitable for publication at Veterinary Sciences after major revision. Introduction, Material and Methods, Results and Discussion sections need to be reviewed carefully by you due to the lack of important information is observed. See below a list of comments/suggestions to be addressed by you before accepting it.

L41 Replace 'MDA' by 'plasma concentrations' and 'lactation' by 'feed additive'.

L46-L49 Explain how stimbiotic acts and provide some examples in animals.

L69-L70 Add references to this statement.

L93 Specify if 3 kg per day were of fresh or dry material.

L113-L116 Describe the instruments and the material used for collection of samples and the equipment managed.

L127-L130 Provide detailed information about the equipment used for measurement and the technique applied (reagents, standards, etc.).

L141 Mention the equipment used.

L142 Add reference to the method applied and mention the equipment.

L143-L144 Add reference to the technique applied and the equipment used.

L146-L152 Describe the statistical tests applied and the models tested.

L205 Provide a table with results related to SCFAs such as acetate, propionate and butyrate. Comment also the results obtained in this section.

L219-L229 Check these sentences. No results were provided in the article concerning to SCFAs measured herein to justify these statements. A justification is needed. It should be based on data and not in speculations.

L250-L261 Justification is needed by adding results from SCFAs measured.

Best regards,

Reviewer.

Author Response

Thank you very much for the great comments that aim to improve the quality of our manuscript. We have modified the manuscript according to your comments.

The modified details were as follows:

 Comments 1: L41 Replace 'MDA' by 'plasma concentrations' and 'lactation' by 'feed additive'.

Response 1: We have replaced 'MDA' by 'plasma concentrations' and 'lactation' by 'feed additive' in Key words of Line 41.

Comments 2: L46-L49 Explain how stimbiotic acts and provide some examples in animals.

Response 2: We have added more details of stimbiotic action in Line 46-49, as follows: “Recent research has reported stimbiotic supplementation enhanced animals’ growth rate and gut health by extracting extra energy from dietary fibre, increasing the utilization of the undigested dietary fibre by promoting fibre-degrading bacterium, breaking down the complex fibre structure into a lower degree of polymerization (DP) fibre, and eventually promoting the production of short chain fatty acids (SCFAs) in the animal’s distal intestine. It was well known that SCFAs, as a main energy source of intestinal epithelial cells (IECs), play an important role on the improvement of the gut integrity and immune function. Short chain fatty acids may also inhibit pathogenic bacteria and reduce the inflammatory response and intestinal diseases, which provide multiple beneficial effects on host health and performance.”

Comments 3: L69-L70 Add references to this statement.

Response 3: We added the references according to your comments.

Comments 4: L93 Specify if 3 kg per day were of fresh or dry material.

Response 4: We have corrected the description “3 kg experimental diet per day”.

Comments 5: L113-L116 Describe the instruments and the material used for collection of samples and the equipment managed.

Response 5: We have described the content in Line 113-116, as follows."The milk samples were collected into sterile tubes and immediately stored at -20°C until further analysis. On d 0 and 28 of lactation, 10 ml of blood was collected from the ear vein of each sow, placed in heparin tubes, and centrifuged at 3000 rpm for 20 minutes by high speed centrifuge (Thermo Fisher, Germany). The plasma was obtained and samples were stored at -20°C for later analysis."

Comments 6: L127-L130 Provide detailed information about the equipment used for measurement and the technique applied (reagents, standards, etc.).

Response 6:  For the microplate spectrophotometer (BioTek, USA) I have mentioned in the last part of 2.5 antioxidant capacities, which included all the optical density (OD) detected equipment. And for technique applied, actually we used ELISA kit for these indicators, it has already provided all the reagents and standards, which we used in this experiment. 

Comments 7: L141 Mention the equipment used.

Response 7: We have added the information, “All the OD values, at different nanometer (nm), in this experiment were conducted on a microplate spectrophotometer (BioTek, USA)".

Comments 8: L142 Add reference to the method applied and mention the equipment.

Response 8: We have added the information in the last part of 2.5 antioxidant capacities, “All the OD values at different nanometer (nm) in this experiment were received by microplate spectrophotometer (BioTek, USA)”

Comments 9: L143-L144 Add reference to the technique applied and the equipment used.

Response 9: We have added the information in the last part of 2.5 antioxidant capacities,  “All the OD values, at different nanometer (nm), in this experiment were conducted on a microplate spectrophotometer (BioTek, USA)".

Comments 10: L146-L152 Describe the statistical tests applied and the models tested.

Response 10: We have updated the description to make it more sense.

Comments 11: L205 Provide a table with results related to SCFAs such as acetate, propionate and butyrate. Comment also the results obtained in this section.

Response 11: Indeed we do not have the date of SCFAs content in this experiment, but in our previous study, we evaluated the performance, feaca microbiome, and SCFAs content in weaned piglet and broiler, which showed very positive results, and indicated better fiber fermentation in distal intestine when fed stimbiotic.  

Comments 12 and 13: L219-L229 Check these sentences. No results were provided in the article concerning to SCFAs measured herein to justify these statements. A justification is needed. It should be based on data and not in speculations. And L250-L261 Justification is needed by adding results from SCFAs measured.

Response 12 and 13: We totally agree with you, it would make more sense if adding the SCFAs content in this experiment, but unfortunately, we lost these data, so we couldn’t provide more evidence on this part. However, for stimbiotic, to be honest, there were many studies have proven that it could increase fiber fermentation in hindgut and enhance the production of SCFAs. That’s why we’d like to discuss it form its MOA and link to our results.

Reviewer 4 Report

Comments and Suggestions for Authors

Your conclusions are too solid - you need to discuss more

There were fewer piglets weaned in the treatment group and therefore it would be expected that the piglets were heavier - which they are

The treatment group weaned what would be too few piglets in todays world and this is not discussed.

Author Response

Thank you very much for the great comments that aim to improve the quality of our manuscript. We have modified the manuscript according to your comments. The more details as follows:

Comments 1: Your conclusions are too solid - you need to discuss more

Response 1: Thank you for the great comments. We have added more information in conclusion part. As follows: “In addition, it also showed that stimbiotic tended to reduce oxidative stress of lactating sows, which indicated stimbiotic could be used as a feed additive in sow diets to improve sow resilience to stressors and piglet performance during lactation period.  “

Comments 2: There were fewer piglets weaned in the treatment group and therefore it would be expected that the piglets were heavier - which they are

Response 2: Thank you for the great comments. We agree that our sow performance seems lower than the commercial farm, which might be due to the imperfect management. However, for the fewer piglets weaned in the treatment group, we should consider the number of born alive at farrowing in treatment group, because it’s already 0.5 piglets lower than the control group. It’s well known that heavier weight of piglet was related to the piglet numbers, while it should be not the main reason since the size of litter in each treatment is affordable for the sows. Actually, milk supply, milk quality and piglet health were related to the higher weight of piglet. In our trial, we found the higher immunoglobulin contents in the colostrum and d14 milk in treatment group, which might be the main reason to increase the piglet weight during lactation period.

Comments 3: The treatment group weaned what would be too few piglets in todays world and this is not discussed.

Response 3: Thank you for the great comments. For the few weaned piglets in the treatment group and the control group, we have mentioned in the above comments, the main reason might be the imperfect management in our trial farm compared to that in the commercial pig farm. And for this trial, we more focused on the evaluation of the effect of stimboitic on sows performance and it was found that higher milk quality and weight of piglet and healthier sow during lactation period. 

Comments 4: NRC 2012 might be less appropriate for modern hyperprolific sows

Response 4: Thanks for the comments. In fact, the sows used in the experimental were Yorkshire ✕Landrace crossed bred from new breed American pigs, which compliant with NRC 2012 standards.

Comments 5: NE would be more appropriate I suggest?

Response 5: Thanks for the comments, we have modified ME to NE in the formulation.

Comments 6: Lysine low for commercial sows in lactation

Response 6: Thanks for the comments. Lysine content for the lactation phase in sow diet was according to NRC standards and used the same level in CT and VP leacation diets to make the consistence.

Comments 7: Line 110: 10 ml in total or from each mammae,

Response 7: Thanks for the comments, we have modified the sentence “A 10-mL sample of the first colostrum of each sow was collected manually from all active mammary glands on one side of the sow within 1 hour during farrowing”。

Comments 8: Age of sows

Response 8: We added the information (at 85 days of gestation) in the Materials and Methods section.

Comments 9: Line 217-218: Need to discuss lower number of piglets less competition around milk supply there are other reasons

Response 9: Thank you for the great comments. We have modified the sentence in Line 217-218 to make it more sense.

Round 2

Reviewer 1 Report

Comments and Suggestions for Authors

The authors have significantly improved the manuscript.

Therefore, I recommend the manuscript for publication in the current form.

Author Response

Dear Reviewer,

      Thanks for the great review. We're very appreciate of that.

       Best Regards,

        Li Jing 

Reviewer 3 Report

Comments and Suggestions for Authors

Dear Authors,

Your manuscript is now suitable for publication in present form.

Yours sincerely,

Reviewer.

Author Response

(The authors gave the same response as above.)

Reviewer 4 Report

Comments and Suggestions for Authors

Interesting and reads much better

My only real comment is Oxytocin should be in IU as there are different concentrations commercially available.

Author Response

Dear Reviewer,

      Thanks for the great comments. we have added more information of Oxyctin using in the materials and methods part. as follows:

"On d 14 of lactation, a 20-mL sample of the milk from each sow was collected approximately 30s after sows were injected with 2 mL (equal to 10 units) of oxytocin by intramuscularly in the hip (Ningbo Sansheng Biotech, Zhejiang, China)"

     Thank you very much. 

      Best Regards,

      Li Jing